# Model-Powered Conditional Independence Test

Rajat Sen[1,*], Ananda Theertha Suresh[2,*], Karthikeyan Shanmugam[3,*], Alexandros G. Dimakis[1], and Sanjay Shakkottai[1]

[1]The University of Texas at Austin
[2]Google, New York
[3]IBM Research, Thomas J. Watson Center

## Abstract

We consider the problem of non-parametric Conditional Independence testing (CI testing) for continuous random variables. Given i.i.d samples from the joint distribution $f(x, y, z)$ of continuous random vectors $X, Y$ and $Z$, we determine whether $X \perp\!\!\!\perp Y | Z$. We approach this by converting the conditional independence test into a classification problem. This allows us to harness very powerful classifiers like gradient-boosted trees and deep neural networks. These models can handle complex probability distributions and allow us to perform significantly better compared to the prior state of the art, for high-dimensional CI testing. The main technical challenge in the classification problem is the need for samples from the conditional product distribution $f^{CI}(x, y, z) = f(x|z)f(y|z)f(z)$ – the joint distribution if and only if $X \perp\!\!\!\perp Y | Z$. – when given access only to i.i.d. samples from the true joint distribution $f(x, y, z)$. To tackle this problem we propose a novel nearest neighbor bootstrap procedure and theoretically show that our generated samples are indeed close to $f^{CI}$ in terms of total variational distance. We then develop theoretical results regarding the generalization bounds for classification for our problem, which translate into error bounds for CI testing. We provide a novel analysis of Rademacher type classification bounds in the presence of non-i.i.d *near-independent* samples. We empirically validate the performance of our algorithm on simulated and real datasets and show performance gains over previous methods.

## 1   Introduction

Testing datasets for Conditional Independence (CI) have significant applications in several statistical/learning problems; among others, examples include discovering/testing for edges in Bayesian networks [15, 27, 7, 9], causal inference [23, 14, 29, 5] and feature selection through Markov Blankets [16, 31]. Given a triplet of random variables/vectors $(X, Y, Z)$, we say that $X$ is conditionally independent of $Y$ given $Z$ (denoted by $X \perp\!\!\!\perp Y | Z$), if the joint distribution $f_{X,Y,Z}(x, y, z)$ factorizes as $f_{X,Y,Z}(x, y, z) = f_{X|Z}(x|z)f_{Y|Z}(y|z)f_Z(z)$. The problem of *Conditional Independence Testing* (CI Testing) can be defined as follows: Given $n$ i.i.d samples from $f_{X,Y,Z}(x, y, z)$, distinguish between the two hypothesis $\mathcal{H}_0 : X \perp\!\!\!\perp Y | Z$ and $\mathcal{H}_1 : X \not\!\perp\!\!\!\perp Y | Z$.

In this paper we propose a data-driven *Model-Powered* CI test. The central idea in a model-driven approach is to convert a statistical testing or estimation problem into a pipeline that utilizes the power of supervised learning models like classifiers and regressors; such pipelines can then leverage recent advances in classification/regression in high-dimensional settings. In this paper, we take such a model-powered approach (illustrated in Fig. 1), which reduces the problem of CI testing to Binary Classification. Specifically, the key steps of our procedure are as follows:

---

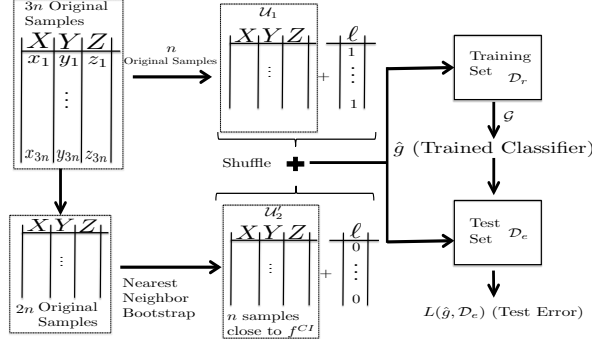

Figure 1: Illustration of our methodology. A part of the original samples are kept aside in $\mathcal{U}_1$. The rest of the samples are used in our nearest neighbor boot-strap to generate a data-set $\mathcal{U}_2'$ which is close to $f^{CI}$ in distribution. The samples are labeled as shown and a classifier is trained on a training set. The test error is measured on a test set there-after. If the test-error is close to $0.5$, then $\mathcal{H}_0$ is not rejected, however if the test error is low then $\mathcal{H}_0$ is rejected.

$(i)$ Suppose we are provided $3n$ i.i.d samples from $f_{X,Y,Z}(x,y,z)$. We keep aside $n$ of these original samples in a set $\mathcal{U}_1$ (refer to Fig. 1). The remaining $2n$ of the original samples are processed through our first module, the *nearest-neighbor bootstrap* (Algorithm 1 in our paper), which produces $n$ simulated samples stored in $\mathcal{U}_2'$. In Section 3, we show that these generated samples in $\mathcal{U}_2'$ are in fact close in total variational distance (defined in Section 3) to the conditionally independent distribution $f^{CI}(x,y,z) \triangleq f_{X|Z}(x|z)f_{Y|Z}(y|z)f_Z(z)$. (Note that only under $\mathcal{H}_0$ does the equality $f^{CI}(.) = f_{X,Y,Z}(.)$ hold; our method generates samples close to $f^{CI}(x,y,z)$ under *both* hypotheses).

$(ii)$ Subsequently, the original samples kept aside in $\mathcal{U}_1$ are labeled 1 while the new samples simulated from the nearest-neighbor bootstrap (in $\mathcal{U}_2'$) are labeled 0. The labeled samples ($\mathcal{U}_1$ with label 1 and $\mathcal{U}_2'$ labeled 0) are aggregated into a data-set $\mathcal{D}$. This set $\mathcal{D}$ is then broken into training and test sets $\mathcal{D}_r$ and $\mathcal{D}_e$ each containing $n$ samples each.

$(iii)$ Given the labeled training data-set (from step $(ii)$), we train powerful classifiers such as gradient boosted trees [6] or deep neural networks [17] which attempt to learn the classes of the samples. If the trained classifier has good accuracy over the test set, then intuitively it means that the joint distribution $f_{X,Y,Z}(.)$ is distinguishable from $f^{CI}$ (note that the generated samples labeled 0 are close in distribution to $f^{CI}$). Therefore, we reject $\mathcal{H}_0$. On the other hand, if the classifier has accuracy close to random guessing, then $f_{X,Y,Z}(.)$ is in fact close to $f^{CI}$, and we fail to reject $\mathcal{H}_0$.

For independence testing (i.e whether $X \perp\!\!\!\perp Y$), classifiers were recently used in [19]. Their key observation was that given i.i.d samples $(X,Y)$ from $f_{X,Y}(x,y)$, if the $Y$ coordinates are randomly permuted then the resulting samples exactly emulate the distribution $f_X(x)f_Y(y)$. Thus the problem can be converted to a two sample test between a subset of the original samples and the other subset which is permuted - Binary classifiers were then harnessed for this two-sample testing; for details see [19]. However, in the case of CI testing we need to emulate samples from $f^{CI}$. This is harder because the permutation of the samples needs to be $Z$ dependent (which can be high-dimensional). One of our key technical contributions is in proving that our nearest-neighbor bootstrap in step $(i)$ achieves this task.

The advantage of this modular approach is that we can harness the power of classifiers (in step $(iii)$ above), which have good accuracies in high-dimensions. Thus, any improvements in the field of binary classification imply an advancement in our CI test. Moreover, there is added flexibility in choosing the best classifier based on domain knowledge about the data-generation process. Finally, our bootstrap is also efficient owing to fast algorithms for identifying nearest-neighbors [24].

## 1.1 Main Contributions

$(i)$ **(Classification based CI testing)** We reduce the problem of CI testing to Binary Classification as detailed in steps $(i)$-$(iii)$ above and in Fig. 1. We simulate samples that are close to $f^{CI}$ through a novel nearest-neighbor bootstrap (Algorithm 1) given access to i.i.d samples from the joint distribution.

The problem of CI testing then reduces to a two-sample test between the original samples in $\mathcal{U}_1$ and $\mathcal{U}'_2$, which can be effectively done by binary classifiers.

$(ii)$ **(Guarantees on Bootstrapped Samples)** As mentioned in steps $(i)$-$(iii)$, if the samples generated by the bootstrap (in $\mathcal{U}'_2$) are close to $f^{CI}$, then the CI testing problem reduces to testing whether the data-sets $\mathcal{U}_1$ and $\mathcal{U}'_2$ are distinguishable from each other. We theoretically justify that this is indeed true. Let $\phi_{X,Y,Z}(x,y,z)$ denote the distribution of a sample produced by Algorithm 1, when it is supplied with $2n$ i.i.d samples from $f_{X,Y,Z}(.)$. In Theorem 1, we prove that $d_{TV}(\phi, f^{CI}) = O(1/n^{1/d_z})$ under appropriate smoothness assumptions. Here $d_z$ is the dimension of $Z$ and $d_{TV}$ denotes total variational distance (Def. 1).

$(iii)$ **(Generalization Bounds for Classification under *near-independence*)** The samples generated from the nearest-neighbor bootstrap do not remain i.i.d but they are *close* to i.i.d. We quantify this property and go on to show generalization risk bounds for the classifier. Let us denote the class of function encoded by the classifier as $\mathcal{G}$. Let $\hat{R}$ denote the probability of error of the optimal classifier $\hat{g} \in \mathcal{G}$ trained on the training set (Fig. 1). We prove that under appropriate assumptions, we have

$$ r_0 - \mathcal{O}(1/n^{1/d_z}) \leq \hat{R} \leq r_0 + \mathcal{O}(1/n^{1/d_z}) + \mathcal{O}\left(\sqrt{V}\left(n^{-1/3} + \sqrt{2^{d_z}/n}\right)\right) $$

with high probability, upto log factors. Here $r_0 = 0.5(1 - d_{TV}(f, f^{CI}))$, $V$ is the VC dimension [30] of the class $\mathcal{G}$. Thus when $f$ is equivalent to $f^{CI}$ ($\mathcal{H}_0$ holds) then the error rate of the classifier is close to 0.5. But when $\mathcal{H}_1$ holds the loss is much lower. We provide a novel analysis of Rademacher complexity bounds [4] under near-independence which is of independent interest.

$(iv)$ **(Empirical Evaluation)** We perform extensive numerical experiments where our algorithm outperforms the state of the art [32, 28]. We also apply our algorithm for analyzing CI relations in the protein signaling network data from the flow cytometry data-set [26]. In practice we observe that the performance with respect to dimension of $Z$ scales much better than expected from our worst case theoretical analysis. This is because powerful binary classifiers perform well in high-dimensions.

## 1.2 Related Work

In this paper we address the problem of non-parametric CI testing when the underlying random variables are continuous. The literature on non-parametric CI testing is vast. We will review some of the recent work in this field that is most relevant to our paper.

Most of the recent work in CI testing are kernel based [28, 32, 10]. Many of these works build on the study in [11], where non-parametric CI relations are characterized using covariance operators for Reproducing Kernel Hilbert Spaces (RKHS) [11]. KCIT [32] uses the partial association of regression functions relating $X$, $Y$, and $Z$. RCIT [28] is an approximate version of KCIT that attempts to improve running times when the number of samples are large. KCIPT [10] is perhaps most relevant to our work. In [10], a specific permutation of the samples is used to simulate data from $f^{CI}$. An expensive linear program needs to be solved in order to calculate the permutation. On the other hand, we use a simple nearest-neighbor bootstrap and further we provide theoretical guarantees about the closeness of the samples to $f^{CI}$ in terms of total variational distance. Finally the two-sample test in [10] is based on a kernel method [3], while we use binary classifiers for the same purpose. There has also been recent work on entropy estimation [13] using nearest neighbor techniques (used for density estimation); this can subsequently be used for CI testing by estimating the conditional mutual information $\mathrm{I}(X; Y | Z)$.

Binary classification has been recently used for two-sample testing, in particular for independence testing [19]. Our analysis of generalization guarantees of classification are aimed at recovering guarantees similar to [4], but in a non-i.i.d setting. In this regard (non-i.i.d generalization guarantees), there has been recent work in proving Rademacher complexity bounds for $\beta$-mixing stationary processes [21]. This work also falls in the category of machine learning reductions, where the general philosophy is to reduce various machine learning settings like multi-class regression [2], ranking [1], reinforcement learning [18], structured prediction [8] to that of binary classification.

## 2 Problem Setting and Algorithms

In this section we describe the algorithmic details of our CI testing procedure. We first formally define our problem. Then we describe our bootstrap algorithm for generating the data-set that mimics samples from $f^{CI}$. We give a detailed pseudo-code for our CI testing process which reduces the problem to that of binary classification. Finally, we suggest further improvements to our algorithm.

**Problem Setting:** The problem setting is that of non-parametric *Conditional Independence (CI)* testing given i.i.d samples from the joint distributions of random variables/vectors [32, 10, 28]. We are given $3n$ i.i.d samples from a continuous joint distribution $f_{X,Y,Z}(x,y,z)$ where $x \in \mathbb{R}^{d_x}, y \in \mathbb{R}^{d_y}$ and $z \in \mathbb{R}^{d_z}$. The goal is to test whether $X \perp\!\!\!\perp Y | Z$ i.e whether $f_{X,Y,Z}(x,y,z)$ factorizes as,

$$f_{X,Y,Z}(x,y,z) = f_{X|Z}(x|z) f_{Y|Z}(y|z) f_Z(z) \triangleq f^{CI}(x,y,z)$$

This is essentially a hypothesis testing problem where: $\mathcal{H}_0 : X \perp\!\!\!\perp Y | Z$ and $\mathcal{H}_1 : X \not\!\perp\!\!\!\perp Y | Z$.

**Note:** For notational convenience, we will drop the subscripts when the context is evident. For instance we may use $f(x|z)$ in place of $f_{X|Z}(x|z)$.

**Nearest-Neighbor Bootstrap:** Algorithm 1 is a procedure to generate a data-set $\mathcal{U}'$ consisting of $n$ samples given a data-set $\mathcal{U}$ of $2n$ i.i.d samples from the distribution $f_{X,Y,Z}(x,y,z)$. The data-set $\mathcal{U}$ is broken into two equally sized partitions $\mathcal{U}_1$ and $\mathcal{U}_2$. Then for each sample in $\mathcal{U}_1$, we find the nearest neighbor in $\mathcal{U}_2$ in terms of the $Z$ coordinates. The $Y$-coordinates of the sample from $\mathcal{U}_1$ are exchanged with the $Y$-coordinates of its nearest neighbor (in $\mathcal{U}_2$); the modified sample is added to $\mathcal{U}'$.

---

**Algorithm 1** DataGen - Given data-set $\mathcal{U} = \mathcal{U}_1 \cup \mathcal{U}_2$ of $2n$ i.i.d samples from $f(x,y,z)$ ($|\mathcal{U}_1| = |\mathcal{U}_2| = n$ ), returns a new data-set $\mathcal{U}'$ having $n$ samples.

1: **function** DATAGEN($\mathcal{U}_1, \mathcal{U}_2, 2n$)
2:     $\mathcal{U}' = \emptyset$
3:     **for** $u$ in $\mathcal{U}_1$ **do**
4:         Let $v = (x', y', z') \in \mathcal{U}_2$ be the sample such that $z'$ is the 1-Nearest Neighbor (1-NN) of $z$ (in $\ell_2$ norm) in the whole data-set $\mathcal{U}_2$, where $u = (x, y, z)$
5:         Let $u' = (x, y', z)$ and $\mathcal{U}' = \mathcal{U}' \cup \{u'\}$.
6:     **end for**
7: **end function**

---

One of our main results is that the samples in $\mathcal{U}'$, generated in Algorithm 1 mimic samples coming from the distribution $f^{CI}$. Suppose $u = (x, y, z) \in \mathcal{U}_1$ be a sample such that $f_Z(z)$ is not too small. In this case $z'$ (the 1-NN sample from $\mathcal{U}_2$) will not be far from $z$. Therefore given a fixed $z$, under appropriate smoothness assumptions, $y'$ will be close to an independent sample coming from $f_{Y|Z}(y|z') \sim f_{Y|Z}(y|z)$. On the other hand if $f_Z(z)$ is small, then $z$ is a rare occurrence and will not contribute adversely.

**CI Testing Algorithm:** Now we introduce our CI testing algorithm, which uses Algorithm 1 along with binary classifiers. The psuedo-code is in Algorithm 2 (Classifier CI Test -CCIT).

---

**Algorithm 2** CCITv1 - Given data-set $\mathcal{U}$ of $3n$ i.i.d samples from $f(x,y,z)$, returns if $X \perp\!\!\!\perp Y | Z$.

1: **function** CCIT($\mathcal{U}, 3n, \tau, \mathcal{G}$)
2:     Partition $\mathcal{U}$ into three disjoint partitions $\mathcal{U}_1, \mathcal{U}_2$ and $\mathcal{U}_3$ of size $n$ each, randomly.
3:     Let $\mathcal{U}'_2 = $ DataGen($\mathcal{U}_2, \mathcal{U}_3, 2n$) (Algorithm 1). Note that $|\mathcal{U}'_2| = n$.
4:     Create Labeled data-set $\mathcal{D} := \{(u, \ell = 1)\}_{u \in \mathcal{U}_1} \cup \{(u', \ell' = 0)\}_{u' \in \mathcal{U}'_2}$
5:     Divide data-set $\mathcal{D}$ into train and test set $\mathcal{D}_r$ and $\mathcal{D}_e$ respectively. Note that $|\mathcal{D}_r| = |\mathcal{D}_e| = n$.
6:     Let $\hat{g} = \text{argmin}_{g \in \mathcal{G}} \hat{L}(g, \mathcal{D}_r) := \frac{1}{|\mathcal{D}_r|} \sum_{(u,\ell) \in \mathcal{D}_r} \mathbb{1}\{g(u) \neq l\}$. This is Empirical Risk Minimization for training the classifier (finding the best function in the class $\mathcal{G}$).
7:     If $\hat{L}(\hat{g}, \mathcal{D}_e) > 0.5 - \tau$, then conclude $X \perp\!\!\!\perp Y | Z$, otherwise, conclude $X \not\!\perp\!\!\!\perp Y | Z$.
8: **end function**

---

In Algorithm 2, the original samples in $\mathcal{U}_1$ and the nearest-neighbor bootstrapped samples in $\mathcal{U}_2'$ should be almost indistinguishable if $\mathcal{H}_0$ holds. However, if $\mathcal{H}_1$ holds, then the classifier trained in Line 6 should be able to easily distinguish between the samples corresponding to different labels. In Line 6, $\mathcal{G}$ denotes the space of functions over which risk minimization is performed in the classifier.

We will show (in Theorem 1) that the variational distance between the distribution of one of the samples in $\mathcal{U}_2'$ and $f^{CI}(x, y, z)$ is very small for large $n$. However, the samples in $\mathcal{U}_2'$ are not exactly i.i.d but *close* to i.i.d. Therefore, in practice for finite $n$, there is a small bias $b > 0$ i.e. $\hat{L}(\hat{g}, \mathcal{D}_e) \sim 0.5 - b$, even when $\mathcal{H}_0$ holds. The threshold $\tau$ needs to be greater than $b$ in order for Algorithm 2 to function. In the next section, we present an algorithm where this bias is corrected.

**Algorithm with Bias Correction:** We present an improved bias-corrected version of our algorithm as Algorithm 3. As mentioned in the previous section, in Algorithm 2, the optimal classifier may be able to achieve a loss slightly less that 0.5 in the case of finite $n$, even when $\mathcal{H}_0$ is true. However, the classifier is expected to distinguish between the two data-sets only based on the $Y, Z$ coordinates, as the joint distribution of $X$ and $Z$ remains the same in the nearest-neighbor bootstrap. The key idea in Algorithm 3 is to train a classifier only using the $Y$ and $Z$ coordinates, denoted by $\hat{g}'$. As before we also train another classier using all the coordinates, which is denoted by $\hat{g}$. The test loss of $\hat{g}'$ is expected to be roughly $0.5 - b$, where $b$ is the bias mentioned in the previous section. Therefore, we can just subtract this bias. Thus, when $\mathcal{H}_0$ is true $\hat{L}(\hat{g}', \mathcal{D}_e') - \hat{L}(\hat{g}, \mathcal{D}_e)$ will be close to 0. However, when $\mathcal{H}_1$ holds, then $\hat{L}(\hat{g}, \mathcal{D}_e)$ will be much lower, as the classifier $\hat{g}$ has been trained leveraging the information encoded in all the coordinates.

---

**Algorithm 3** CCITv2 - Given data-set $\mathcal{U}$ of $3n$ i.i.d samples, returns whether $X \perp\!\!\!\perp Y | Z$.

---

1: **function** CCIT($\mathcal{U}, 3n, \tau, \mathcal{G}$)
2:     Perform Steps 1-5 as in Algorithm 2.
3:     Let $\mathcal{D}_r' = \{((y, z), \ell)\}_{(u=(x,y,z),\ell)\in\mathcal{D}_r}$. Similarly, let $\mathcal{D}_e' = \{((y, z), \ell)\}_{(u=(x,y,z),\ell)\in\mathcal{D}_e}$. These are the training and test sets without the $X$-coordinates.
4:     Let $\hat{g} = \text{argmin}_{g\in\mathcal{G}} \hat{L}(g, \mathcal{D}_r) := \frac{1}{|\mathcal{D}_r|} \sum_{(u,\ell)\in\mathcal{D}_r} \mathbb{1}\{g(u) \neq l\}$. Compute test loss: $\hat{L}(\hat{g}, \mathcal{D}_e)$.
5:     Let $\hat{g}' = \text{argmin}_{g\in\mathcal{G}} \hat{L}(g, \mathcal{D}_r') := \frac{1}{|\mathcal{D}_r'|} \sum_{(u,\ell)\in\mathcal{D}_r'} \mathbb{1}\{g(u) \neq l\}$. Compute test loss: $\hat{L}(\hat{g}', \mathcal{D}_e')$.
6:     If $\hat{L}(\hat{g}, \mathcal{D}_e) < \hat{L}(\hat{g}', \mathcal{D}_e') - \tau$, then conclude $X \not\!\perp\!\!\!\perp Y | Z$, otherwise, conclude $X \perp\!\!\!\perp Y | Z$.
7: **end function**

---

## 3 Theoretical Results

In this section, we provide our main theoretical results. We first show that the distribution of any one of the samples generated in Algorithm 1 closely resemble that of a sample coming from $f^{CI}$. This result holds for a broad class of distributions $f_{X,Y,Z}(x, y, z)$ which satisfy some smoothness assumptions. However, the samples generated by Algorithm 1 ($\mathcal{U}_2$ in the algorithm) are not exactly i.i.d but *close* to i.i.d. We quantify this and go on to show that empirical risk minimization over a class of classifier functions generalizes well using these samples. Before, we formally state our results we provide some useful definitions.

**Definition 1.** *The **total variational distance** between two continuous probability distributions $f(.)$ and $g(.)$ defined over a domain $\mathcal{X}$ is, $d_{TV}(f, g) = \sup_{p\in\mathcal{B}} |\mathbb{E}_f[p(X)] - \mathbb{E}_g[p(X)]|$ where $\mathcal{B}$ is the set of all measurable functions from $\mathcal{X} \to [0, 1]$. Here, $\mathbb{E}_f[.]$ denotes expectation under distribution $f$.*

We first prove that the distribution of any one of the samples generated in Algorithm 1 is close to $f^{CI}$ in terms of total variational distance. We make the following assumptions on the joint distribution of the original samples i.e. $f_{X,Y,Z}(x, y, z)$:

**Smoothness assumption on** $f(y|z)$**:** We assume a smoothness condition on $f(y|z)$, that is a generalization of boundedness of the max. eigenvalue of Fisher Information matrix of $y$ w.r.t $z$.

**Assumption 1.** *For $z \in \mathbb{R}^{d_z}$, a such that $\|a - z\|_2 \leq \epsilon_1$, the generalized curvature matrix $\mathbf{I}_a(z)$ is,*

$$\mathbf{I}_a(z)_{ij} = \left( \frac{\partial^2}{\partial z_i' \partial z_j'} \int \log \frac{f(y|z)}{f(y|z')} f(y|z) dy \right) \Bigg|_{z'=a} = \mathbb{E} \left[ -\frac{\delta^2 \log f(y|z')}{\delta z_i' \delta z_j'} \Big|_{z'=a} \Bigg| Z = z \right] \quad (1)$$

*We require that for all $z \in \mathbb{R}^{d_z}$ and all $a$ such that $\|a - z\|_2 \leq \epsilon_1$, $\lambda_{max}(\mathbf{I}_a(z)) \leq \beta$. Analogous assumptions have been made on the Hessian of the density in the context of entropy estimation [12].*

**Smoothness assumptions on $f(z)$:** We assume some smoothness properties of the probability density function $f(z)$. The smoothness assumptions (in Assumption 2) is a subset of the assumptions made in [13] (Assumption 1, Page 5) for entropy estimation.

**Definition 2.** *For any $\delta > 0$, we define $G(\delta) = \mathbb{P}(f(Z) \leq \delta)$. This is the probability mass of the distribution of $Z$ in the areas where the p.d.f is less than $\delta$.*

**Definition 3.** *(Hessian Matrix) Let $H_f(z)$ denote the Hessian Matrix of the p.d.f $f(z)$ with respect to $z$ i.e $H_f(z)_{ij} = \partial^2 f(z)/\partial z_i \partial z_j$, provided it is twice continuously differentiable at $z$.*

**Assumption 2.** *The probability density function $f(z)$ satisfies the following:*

*(1) $f(z)$ is twice continuously differentiable and the Hessian matrix $H_f$ satisfies $\|H_f(z)\|_2 \leq c_{d_z}$ almost everywhere, where $c_{d_z}$ is only dependent on the dimension.*

*(2) $\int f(z)^{1-1/d} dz \leq c_3, \ \forall d \geq 2$ where $c_3$ is a constant.*

**Theorem 1.** *Let $(X, Y', Z)$ denote a sample in $\mathcal{U}_2'$ produced by Algorithm 1 by modifying the original sample $(X, Y, Z)$ in $\mathcal{U}_1$, when supplied with $2n$ i.i.d samples from the original joint distribution $f_{X,Y,Z}(x, y, z)$. Let $\phi_{X,Y,Z}(x, y, z)$ be the distribution of $(X, Y', Z)$. Under smoothness assumptions (1) and (2), for any $\epsilon < \epsilon_1$, $n$ large enough, we have:*

$$d_{TV}(\phi, f^{CI}) \leq b(n)$$

$$\triangleq \frac{1}{2} \sqrt{\frac{\beta}{4} \frac{c_3 * 2^{1/d_z} \Gamma(1/d_z)}{(n\gamma_{d_z})^{1/d_z} d_z} + \frac{\beta \epsilon G\left(2c_{d_z} \epsilon^2\right)}{4}} + \exp\left(-\frac{1}{2} n\gamma_{d_z} c_{d_z} \epsilon^{d_z+2}\right) + G\left(2c_{d_z} \epsilon^2\right).$$

*Here, $\gamma_d$ is the volume of the unit radius $\ell_2$ ball in $\mathbb{R}^d$.*

Theorem 1 characterizes the variational distance of the distribution of a sample generated in Algorithm 1 with that of the conditionally independent distribution $f^{CI}$. We defer the proof of Theorem 1 to Appendix A. Now, our goal is to characterize the misclassification error of the trained classifier in Algorithm 2 under both $\mathcal{H}_0$ and $\mathcal{H}_1$. Consider the distribution of the samples in the data-set $\mathcal{D}_r$ used for classification in Algorithm 2. Let $q(x, y, z|\ell = 1)$ be the marginal distribution of each sample with label 1. Similarly, let $q(x, y, z|\ell = 0)$ denote the marginal distribution of the label 0 samples. Note that under our construction,

$$q(x, y, z|\ell = 1) = f_{X,Y,Z}(x, y, z) = \begin{cases} f^{CI}(x, y, z) & \text{if } \mathcal{H}_0 \text{ holds} \\ \neq f^{CI}(x, y, z) & \text{if } \mathcal{H}_1 \text{ holds} \end{cases}$$

$$q(x, y, z|\ell = 0) = \phi_{X,Y,Z}(x, y, z) \quad (2)$$

where $\phi_{X,Y,Z}(x, y, z)$ is as defined in Theorem 1.

Note that even though the marginal of each sample with label 0 is $\phi_{X,Y,Z}(x, y, z)$ (Equation (2)), they are not exactly i.i.d owing to the nearest neighbor bootstrap. We will go on to show that they are actually *close* to i.i.d and therefore classification risk minimization generalizes similar to the i.i.d results for classification [4]. First, we review standard definitions and results from classification theory [4].

**Ideal Classification Setting:** We consider an *ideal* classification scenario for CI testing and in the process define standard quantities in learning theory. Recall that $\mathcal{G}$ is the set of classifiers under consideration. Let $\tilde{q}$ be our *ideal* distribution for $q$ given by $\tilde{q}(x, y, z|\ell = 1) = f_{X,Y,Z}(x, y, z)$, $\tilde{q}(x, y, z|\ell = 0) = f^{CI}_{X,Y,Z}(x, y, z)$ and $\tilde{q}(\ell = 1) = \tilde{q}(\ell = 0) = 0.5$. In other words this is the ideal classification scenario for testing CI. Let $L(g(u), \ell)$ be our **loss function** for a classifying function $g \in \mathcal{G}$, for a sample $u \triangleq (x, y, z)$ with true label $\ell$. In our algorithms the loss function is the $0 - 1$ loss, but our results hold for any bounded loss function s.t. $|L(g(u), \ell)| \leq |L|$. For a distribution $\tilde{q}$

and a classifier $g$ let $R_{\tilde{q}}(g) \triangleq \mathbb{E}_{u,\ell \sim \tilde{q}}[L(g(u), \ell)]$ be the **expected risk** of the function $g$. The **risk optimal classifier** $g_{\tilde{q}}^*$ under $\tilde{q}$ is given by $g_{\tilde{q}}^* \triangleq \arg\min_{g \in \mathcal{G}} R_{\tilde{q}}(g)$. Similarly for a set of samples $S$ and a classifier $g$, let $R_S(g) \triangleq \frac{1}{|S|} \sum_{u,\ell \in S} L(g(u), \ell)$ be the **empirical risk** on the set of samples. We define $g_S$ as the classifier that **minimizes the empirical loss** on the observed set of samples $S$ that is, $g_S \triangleq \arg\min_{g \in \mathcal{G}} R_S(g)$.

If the samples in $S$ are generated independently from $\tilde{q}$, then standard results from the learning theory states that with probability $\geq 1 - \delta$,

$$R_{\tilde{q}}(g_S) \leq R_{\tilde{q}}(g_{\tilde{q}}^*) + C\sqrt{\frac{V}{n}} + \sqrt{\frac{2\log(1/\delta)}{n}}, \tag{3}$$

where $V$ is the VC dimension [30] of the classification model, $C$ is an universal constant and $n = |S|$.

**Guarantees under near-independent samples:** Our goal is to prove a result like (3), for the classification problem in Algorithm 2. However, in this case we do not have access to i.i.d samples because the samples in $\mathcal{U}_2'$ do not remain independent. We will see that they are close to independent in some sense. This brings us to one of our main results in Theorem 2.

**Theorem 2.** *Assume that the joint distribution $f(x, y, z)$ satisfies the conditions in Theorem 1. Further assume that $f(z)$ has a bounded Lipschitz constant. Consider the classifier $\hat{g}$ in Algorithm 2 trained on the set $\mathcal{D}_r$. Let $S = \mathcal{D}_r$. Then according to our definition $g_S = \hat{g}$. For $\epsilon > 0$ we have:*

*(i)* $R_q(g_S) - R_q(g_q^*) \leq \gamma_n$

$$\triangleq C|L|\left(\left(\sqrt{V} + \sqrt{\log\frac{1}{\delta}}\right)\left(\left(\frac{\log(n/\delta)}{n}\right)^{1/3} + \sqrt{\frac{4^{d_z}\log(n/\delta) + o_n(1/\epsilon)}{n}}\right) + G(\epsilon)\right),$$

*with probability at least $1 - 8\delta$. Here $V$ is the V.C. dimension of the classification function class, $G$ is as defined in Def. 2, $C$ is an universal constant and $|L|$ is the bound on the absolute value of the loss.*

*(ii) Suppose the loss is $L(g(u), \ell) = \mathbb{1}_{g(u) \neq \ell}$ (s.t $|L| \leq 1$). Further suppose the class of classifying functions is such that $R_q(g_q^*) \leq r_0 + \eta$. Here, $r_0 \triangleq 0.5(1 - d_{TV}(q(x, y, z|1), q(x, y, z|0)))$ is the risk of the Bayes optimal classifier when $q(\ell = 1) = q(\ell = 0)$. This is the best loss that any classifier can achieve for this classification problem [4]. Under this setting, w.p at least $1 - 8\delta$ we have:*

$$\frac{1}{2}\left(1 - d_{TV}(f, f^{CI})\right) - \frac{b(n)}{2} \leq R_q(g_S) \leq \frac{1}{2}\left(1 - d_{TV}(f, f^{CI})\right) + \frac{b(n)}{2} + \eta + \gamma_n$$

*where $b(n)$ is as defined in Theorem 1.*

We prove Theorem 2 as Theorem 3 and Theorem 4 in the appendix. In part $(i)$ of the theorem we prove that generalization bounds hold even when the samples are not exactly i.i.d. Intuitively, consider two sample inputs $u_i, u_j \in \mathcal{U}_1$, such that corresponding $Z$ coordinates $z_i$ and $z_j$ are far away. Then we expect the resulting samples $u_i'$ and $u_j'$ (in $\mathcal{U}_2'$) to be nearly-independent. By carefully capturing this notion of spatial near-independence, we prove generalization errors in Theorem 3. Part $(ii)$ of the theorem essentially implies that the error of the trained classifier will be close to $0.5$ (l.h.s) when $f \sim f^{CI}$ (under $\mathcal{H}_0$). On the other hand under $\mathcal{H}_1$ if $d_{TV}(f, f^{CI}) > 1 - \gamma$, the error will be less than $0.5(\gamma + b(n)) + \gamma_n$ which is small.

## 4 Empirical Results

In this section we provide empirical results comparing our proposed algorithm and other state of the art algorithms. The algorithms under comparison are: $(i)$ CCIT - Algorithm 3 in our paper where we use XGBoost [6] as the classifier. In our experiments, for each data-set we boot-strap the samples and run our algorithm $B$ times. The results are averaged over $B$ bootstrap runs[1]. $(ii)$ KCIT - Kernel CI test from [32]. We use the Matlab code available online. $(iii)$ RCIT - Randomized CI Test from [28]. We use the R package that is publicly available.

## 4.1 Synthetic Experiments

We perform the synthetic experiments in the regime of *post-nonlinear noise* similar to [32]. In our experiments $X$ and $Y$ are dimension 1, and the dimension of $Z$ scales (motivated by causal settings and also used in [32, 28]). $X$ and $Y$ are generated according to the relation $G(F(Z) + \eta)$ where $\eta$ is a noise term and $G$ is a non-linear function, when the $\mathcal{H}_0$ holds. In our experiments, the data is generated as follows: $(i)$ when $X \perp\!\!\!\perp Y|Z$, then each coordinate of $Z$ is a Gaussian with unit mean and variance, $X = \cos(a^T Z + \eta_1)$ and $Y = \cos(b^T Z + \eta_2)$. Here, $a, b \in \mathbb{R}^{d_z}$ and $\|a\| = \|b\| = 1$. $a,b$ are fixed while generating a single dataset. $\eta_1$ and $\eta_2$ are zero-mean Gaussian noise variables, which are independent of everything else. We set $Var(\eta_1) = Var(\eta_2) = 0.25$. $(ii)$ when $X \not\perp\!\!\!\perp Y|Z$, then everything is identical to $(i)$ except that $Y = \cos(b^T Z + cX + \eta_2)$ for a randomly chosen constant $c \in [0, 2]$.

In Fig. 2a, we plot the performance of the algorithms when the dimension of $Z$ scales. For generating each point in the plot, 300 data-sets were generated with the appropriate dimensions. Half of them are according to $\mathcal{H}_0$ and the other half are from $\mathcal{H}_1$ Then each of the algorithms are run on these data-sets, and the ROC AUC (Area Under the Receiver Operating Characteristic curve) score is calculated from the true labels (CI or not CI) for each data-set and the predicted scores. We observe that the accuracy of CCIT is close to 1 for dimensions upto 70, while all the other algorithms do not scale as well. In these experiments the number of bootstraps per data-set for CCIT was set to $B = 50$. We set the threshold in Algorithm 3 to $\tau = 1/\sqrt{n}$, which is an upper-bound on the expected variance of the test-statistic when $\mathcal{H}_0$ holds.

## 4.2 Flow-Cytometry Dataset

We use our CI testing algorithm to verify CI relations in the protein network data from the flow-cytometry dataset [26], which gives expression levels of 11 proteins under various experimental conditions. The ground truth causal graph is not known with absolute certainty in this data-set, however this dataset has been widely used in the causal structure learning literature. We take three popular learned causal structures that are recovered by causal discovery algorithms, and we verify CI relations assuming these graphs to be the ground truth. The three graph are: $(i)$ consensus graph from [26] (Fig. 1(a) in [22]) $(ii)$ reconstructed graph by Sachs et al. [26] (Fig. 1(b) in [22]) $(iii)$ reconstructed graph in [22] (Fig. 1(c) in [22]).

For each graph we generate CI relations as follows: for each node $X$ in the graph, identify the set $Z$ consisting of its parents, children and parents of children in the causal graph. Conditioned on this set $Z$, $X$ is independent of every other node $Y$ in the graph (apart from the ones in $Z$). We use this to create all CI conditions of these types from each of the three graphs. In this process we generate over 60 CI relations for each of the graphs. In order to evaluate false positives of our algorithms, we also need relations such that $X \not\perp\!\!\!\perp Y|Z$. For, this we observe that if there is an edge between two nodes, they are never CI given any other conditioning set. For each graph we generate 50 such non-CI relations, where an edge $X \leftrightarrow Y$ is selected at random and a conditioning set of size 3 is randomly selected from the remaining nodes. We construct 50 such negative examples for each graph. In Fig. 2, we display the performance of all three algorithms based on considering each of the three graphs as ground-truth. The algorithms are given access to observational data for verifying CI and non-CI relations. In Fig. 2b we display the ROC plot for all three algorithms for the data-set generated by considering graph $(ii)$. In Table 2c we display the ROC AUC score for the algorithms for the three graphs. It can be seen that our algorithm outperforms the others in all three cases, even when the dimensionality of $Z$ is fairly low (less than 10 in all cases). An interesting thing to note is that the edges (pkc-raf), (pkc-mek) and (pka-p38) are there in all the three graphs. However, all three CI testers CCIT, KCIT and RCIT are fairly confident that these edges should be absent. These edges may be discrepancies in the ground-truth graphs and therefore the ROC AUC of the algorithms are lower than expected.

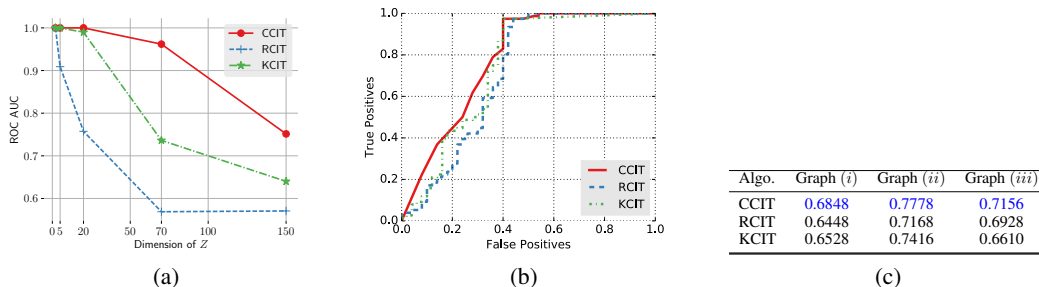

| | | (a) | | | (b) | | | (c) |

Figure 2: In (a) we plot the performance of CCIT, KCIT and RCIT in the post-nonlinear noise synthetic data. In generating each point in the plots, 300 data-sets are generated where half of them are according to $\mathcal{H}_0$ while the rest are according to $\mathcal{H}_1$. The algorithms are run on each of them, and the ROC AUC score is plotted. In $(a)$ the number of samples $n = 1000$, while the dimension of $Z$ varies. In $(b)$ we plot the ROC curve for all three algorithms based on the data from Graph $(ii)$ for the flow-cytometry dataset. The ROC AUC score for each of the algorithms are provided in $(c)$, considering each of the three graphs as ground-truth.

## 5   Conclusion

In this paper we present a model-powered approach for CI tests by converting it into binary classification, thus empowering CI testing with powerful supervised learning tools like gradient boosted trees. We provide an efficient nearest-neighbor bootstrap which makes the reduction to classification possible. We provide theoretical guarantees on the bootstrapped samples, and also risk generalization bounds for our classification problem, under non-i.i.d near independent samples. In conclusion we believe that model-driven data dependent approaches can be extremely useful in general statistical testing and estimation problems as they enable us to use powerful supervised learning tools.

### Acknowledgments

This work is partially supported by NSF grants CNS 1320175, NSF SaTC 1704778, ARO grants W911NF-17-1-0359, W911NF-16-1-0377 and the US DoT supported D-STOP Tier 1 University Transportation Center.

## Footnotes

[1]The python package for our implementation can be found here (https://github.com/rajatsen91/CCIT).

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
