[Supplementary Material]

# A  Guarantees on Bootstrapped Samples

In this section we prove that the samples generated in Algorithm 1, through the nearest neighbor bootstrap, are close to samples generated from $f^{CI}(x, y, z) = f_{X|Z}(x|z)f_{Y|Z}(y|z)f_Z(z)$. The *closeness* is characterized in terms of total variational distance as in Theorem 1. Suppose $2n$ i.i.d samples from distribution $f(x, y, z)$ are supplied to Algorithm 1. Consider a typical sample $(X, Y, Z) \sim f(x, y, z)$, which is modified to produce a typical sample in $\mathcal{U}_2'$ (refer to Algorithm 1) denoted by $(X, Y', Z)$. Here, $Y'$ are the $Y$-coordinates of a sample $(X', Y', Z')$ in $\mathcal{U}_2$ such that $Z'$ is the nearest neighbor of $Z$. Let us denote the marginal distribution of a typical sample in $\mathcal{U}_2'$ by $\phi_{X,Y,Z}(x, y, z)$, i.e $(X, Y', Z) \sim \phi_{X,Y,Z}(x, y, z)$. Now we are at a position to prove Theorem 1.

*Proof of Theorem 1.* Let $f_{Z'|z}(z')$ denote the conditional p.d.f of the variable $Z'$ (that is the nearest neighbor of sample $Z$ in $\mathcal{U}_2$), conditioned on $Z = z$. Therefore, the distribution of the new-sample is given by,

$$\phi_{X,Y,Z}(x, y, z) = f_{X|Z}(x|z)f_Z(z) \int f_{Y|Z}(y|z')f_{Z'|z}(z')dz'. \tag{4}$$

We want to bound the total variational distance between $\phi_{X,Y,Z}(x, y, z)$ and $f^{CI}_{X,Y,Z}(x, y, z)$. We have the following chain:

$$2 * d_{TV}(\phi, f^{CI}) = \int_{x,y,z} \left| f_{X|Z}(x|z)f_{Y|Z}(y|z)f_Z(z) - f_{X|Z}(x|z)f_Z(z) \int f_{Y|Z}(y|z')f_{Z'|z}(z')dz' \right| dxdydz$$

$$= \int_{x,y,z} f_{X|Z}(x|z)f_{Y|Z}(y|z)f_Z(z) \left| \int \left( 1 - \frac{f_{Y|Z}(y|z')}{f_{Y|Z}(y|z)} \right) f_{Z'|z}(z')dz' \right| dxdydz$$

$$\leq \int_{x,y,z} f_{X|Z}(x|z)f_{Y|Z}(y|z)f_Z(z) \int \left| 1 - \frac{f_{Y|Z}(y|z')}{f_{Y|Z}(y|z)} \right| f_{Z'|z}(z')dz' dxdydz$$

$$= \int_{x,z,z'} f_{X|Z}(x|z)f_Z(z)f_{Z'|z}(z') \left( \int \left| f_{Y|Z}(y|z) - f_{Y|Z}(y|z') \right| dy \right) dz' dxdz$$

$$\leq \int_{x,z,\|z'-z\|_2 \leq \epsilon} f_{X|Z}(x|z)f_Z(z)f_{Z'|z}(z') \left( \int \left| f_{Y|Z}(y|z) - f_{Y|Z}(y|z') \right| dy \right) dz' dxdz +$$

$$2 \int_{x,z,\|z'-z\|_2 > \epsilon} f_{X|Z}(x|z)f_Z(z)f_{Z'|z}(z')dz' dxdz$$

$$\leq \int_{x,z,\|z'-z\|_2 \leq \epsilon} f_{X|Z}(x|z)f_Z(z)f_{Z'|z}(z') \left( \int \left| f_{Y|Z}(y|z) - f_{Y|Z}(y|z') \right| dy \right) dz' dxdz +$$

$$2 * \mathbb{P}(\|z'-z\|_2 > \epsilon) \tag{5}$$

By Pinsker's inequality, we have:

$$\int_y \left| f_{Y|Z}(y|z) - f_{Y|Z}(y|z') \right| dy \leq \sqrt{\frac{1}{2} \int_y \log \frac{f(y|z)}{f(y|z')} f(y|z)dy} \tag{6}$$

By Taylor's expansion with second-order residual, we have:

$$\int \log \frac{f(y|z)}{f(y|z')} f(y|z)dy = \frac{1}{2}(z'-z)^T \mathbf{I}_a(z)(z'-z) \tag{7}$$

for some $a = \lambda z + (1-\lambda)z'$ where $0 \leq \lambda \leq 1$.

Under Assumption 1 and $\epsilon < \epsilon_1$ we have,

$$(z' - z)^T \mathbf{I}_a(z)(z' - z) \leq \beta \|z' - z\|_2^2. \tag{8}$$

Then, (8), (7), (6) and (5) imply:

$$2 * d_{TV}(\phi, f^{CI}) \leq \sqrt{\frac{\beta}{4} \mathbb{E}[\|z' - z\|_2 \mathbb{1}_{\|z'-z\|_2 \leq \epsilon}]} + 2\mathbb{P}\left(\|z' - z\|_2 > \epsilon\right) \tag{9}$$

We now bound both terms separately. Let $Z_1, Z_2..., Z_n$ be distributed i.i.d according to $f(z)$. Then, $f_{Z'|z}(\cdot)$ is the pdf of the nearest neighbor of $z$ among $Z_1, \ldots, Z_n$.

### A.0.1 Bounding the first term

In this section we will use $d$ in place of $d_z$ for notational simplicity. Let $\gamma_d$ be the volume of the unit $\ell_2$ ball in dimension $d$. Let $S = \{z : f(z) \geq 2 * c_d \epsilon^2\}$. This implies, that for $z \in S$:

$$f(z) - c_d \epsilon^2 \geq f(z)/2 \tag{10}$$

Let $Z' = \operatorname{argmin}_{Z_1, Z_2..., Z_n} \|Z_i - z\|_2$ be the random variable which is the nearest neighbor to a point $z$ among $n$ i.i.d samples $Z_i$ drawn from the distribution whose pdf is $f(z)$ that satisfies assumption 2. Let $r(z) = ||z - z'||_2$. Let $F(r)$ be the CDF of the random variable $R$. Since $R$ is a non-negative random variable,

$$\mathbb{E}_R[r(z)\mathbb{1}_{r \leq \epsilon}] = \int_{r \leq \epsilon} r dF(r) = [rF(r)]_0^\epsilon - \int_{r \leq \epsilon} F(r)dr \leq \int_{r \leq \epsilon} P(R > r)dr \tag{11}$$

For any $r \leq \epsilon$, observe that

$$\begin{aligned}
\Pr(R > r) &= \Pr(\nexists i : z_i \in B(z, r)) \\
&= (1 - \Pr(Z \in B(z, r)))^n \\
&\leq \exp(-n\Pr(Z \in B(z, r)))
\end{aligned} \tag{12}$$

We have the following chain to bound $\Pr(Z \in B(z, r))$. Let $a = \lambda z + (1 - \lambda)t$.

$$\begin{aligned}
|\Pr(Z \in B(z, r)) - f(z)\gamma_d r^d| &\leq \left| \int_{t \in B(z,r)} (f(t) - f(z))dt \right| \\
&= \left| \int_{t \in B(z,r)} (\nabla f(z)^T.(t - z) + (t - z)^T H_f(a)(t - z))dt \right| \\
&\leq \max_{t \in B(z,r)} \|H_f(t)\|_2 \int_{t \in B(z,r)} \|t - z\|_2^2 dt \\
&\leq c_d \gamma_d r^{d+2}
\end{aligned} \tag{13}$$

By putting together (11),(12) and(13), we have:

$$\begin{aligned}
\mathbb{E}_R[r(z)\mathbb{1}_{r \leq \epsilon}] &\leq \mathbb{1}_{z \in S} \left( \int_{r \leq \epsilon} e^{-n\gamma_d r^d(f(z) - c_d r^2)} dr \right) + \epsilon \mathbb{1}_{z \in S^c} \\
&\leq \mathbb{1}_{z \in S} \left( \int_{r \leq \epsilon} e^{-\frac{1}{2} n\gamma_d f(z) r^d} dr \right) + \epsilon \mathbb{1}_{z \in S^c} \\
&\leq \mathbb{1}_{z \in S} \left( \int_{t \leq \frac{1}{2} n\gamma_d f(z)\epsilon^d} \frac{e^{-t}}{d[\frac{1}{2}n\gamma_d f(z)]^{1/d}} t^{-1+1/d} dt \right) + \epsilon \mathbb{1}_{z \in S^c} \\
&\leq \mathbb{1}_{z \in S} \left( \frac{2^{1/d}}{d(n\gamma_d f(z))^{1/d}} \Gamma(1/d) \right) + \epsilon \mathbb{1}_{z \in S^c}
\end{aligned}$$

Therefore, the first term in bounded by:

$$\mathbb{E}[\|z' - z\|_2 \mathbb{1}_{\|z'-z\|_2 \le \epsilon}] \le \mathbb{E}_Z\left[\mathbb{E}_R[r(z)\mathbb{1}_{r \le \epsilon}]\right] \tag{14}$$

$$\le \mathbb{E}_Z\left[\frac{2^{1/d}}{d(n\gamma_d f(z))^{1/d}}\Gamma(1/d) + \epsilon \mathbb{1}_{z \in S^c}\right] \tag{15}$$

$$\le \frac{c_3 * 2^{1/d}}{(n\gamma_d)^{1/d}d}\Gamma(1/d) + \epsilon * G\left(2c_d\epsilon^2\right) \tag{16}$$

$$\tag{17}$$

### A.0.2 Bounding the second term

We now bound the second term as follows:

$$\Pr(\|z - z'\|_2 > \epsilon) \le \mathbb{E}_Z[\Pr(R > \epsilon)]$$

$$\le \mathbb{E}_Z\left[\mathbb{1}_{z \in S}\exp\left(-\frac{n\gamma_d f(z)\epsilon^d}{2}\right)\right] + \Pr(z \in S^c)$$

$$\le \exp\left(-n\gamma_d c_d \epsilon^{d+2}/2\right) + G\left(2c_d\epsilon^2\right)$$

Substituting in (9), we have:

$$2 * d_{TV}(g, f^{CI}) \le \sqrt{\frac{\beta}{4}\frac{c_3 * 2^{1/d}\Gamma(1/d)}{(n\gamma_d)^{1/d}d} + \frac{\beta\epsilon G\left(2c_d\epsilon^2\right)}{4}} + 2\exp\left(-n\gamma_d c_d \epsilon^{d+2}/2\right) + 2G\left(2c_d\epsilon^2\right)$$

Substitute $d_z$ in place of $d$ to recover Theorem 1.

$\square$

## B  Generalization Error Bounds on Classification

In this section, we will prove generalization error bounds for our classification problem in Algorithm 2. Note the samples in $\mathcal{U}_2'$ are not i.i.d, so standard risk bounds do not hold. We will leverage a spatial near independence property to provide generalization bounds under non-i.i.d samples. In what follows, we will prove the results for any bounded loss function $L(g(u), \ell) \le |L|$. Let $S \triangleq \mathcal{D}_r$ i.e., the set of training samples. For $1 \le i \le 3$, let $\mathcal{Z}_i \triangleq \{z : (x, y, z) \in \mathcal{U}_i\}$. Let $\mathcal{Z} \triangleq \mathcal{Z}_1 \cup \mathcal{Z}_2$. Observe that

$$R_q(g_S) \le R_q(g_q^*) + 2\sup_{g \in \mathcal{G}}(R_S(g) - R_q(g)), \tag{18}$$

and hence in the rest of the section we upper bound $\sup_{g \in \mathcal{G}}(R_S(g) - R_q(g))$. To this end, we define conditional risk $R_S(q|\mathcal{Z})$ as

$$R_S(g|\mathcal{Z}) \triangleq \frac{1}{n}\sum_{(u,\ell) \in S}\mathbb{E}[L(g(u), \ell)|\mathcal{Z}].$$

By triangle inequality,

$$\sup_{g \in \mathcal{G}}(R_S(g) - R_q(g)) \le \sup_{g \in \mathcal{G}}(R_S(g) - R_S(g|\mathcal{Z})) + \sup_{g \in \mathcal{G}}(R_S(g|\mathcal{Z}) - R_q(g)). \tag{19}$$

We first bound the second term in the right hand side of Equation (19) in the next lemma.

**Lemma 1.** *With probability at least $1 - \delta$,*

$$\sup_{g \in \mathcal{G}}(R_S(g|\mathcal{Z}) - R_q(g)) \le |L|C\sqrt{\frac{V}{n}} + |L|\sqrt{\frac{2\log(1/\delta)}{n}},$$

*where $V$ is the VC dimension of the classification model.*

*Proof.* For a sample $u = (x, y, z)$, observe that $\mathcal{Z} \to z \to u$ forms a Markov chain. Hence,

$$\mathbb{E}[L(g(u), \ell)|\mathcal{Z}] = \mathbb{E}[L(g(u), \ell)|Z].$$

Let

$$h(Z) \triangleq \mathbb{E}[L(g(u), \ell)|Z] \tag{20}$$

Hence,

$$R_S(g|\mathcal{Z}) - R_q(g) = \frac{1}{n} \sum_{z \in \mathcal{Z}} h(Z) - \mathbb{E}_q[h(Z)].$$

The above term is the average of $n$ independent random variables $h(Z)$ and hence we can apply standard tools from learning theory [4] to obtain

$$\sup_{g \in \mathcal{G}}(R_S(g|\mathcal{Z}) - R_q(g)) \leq |L|C\sqrt{\frac{V_h}{n}} + |L|\sqrt{\frac{2\log(1/\delta)}{n}},$$

where $V_h$ is the VC dimension of the class of models of $h$. The lemma follows from the fact that VC dimension of $h$ is smaller than the VC dimension of the underlying classification model. $\square$

We next bound the first term in the RHS of Equation (19). Proof is given in Appendix B.1.

**Lemma 2.** *Let $\epsilon > 0$. If the Hessian of the density $f(z)$ and the Lipscitz constant of the same is bounded, then with probability at least $1 - 7\delta$,*

$$\sup_{g \in \mathcal{G}}(R_S(g) - R_n(g|\mathcal{Z})) \leq |L|\left(\sqrt{V} + \sqrt{\log\frac{1}{\delta}}\right)\left(\left(\frac{\log(n/\delta)}{n}\right)^{1/3} + \sqrt{\frac{4^d \log(n/\delta) + o_n(1/\epsilon)}{n}}\right)$$
$$+ |L|G(\epsilon). \tag{21}$$

Lemmas 1 and 2, together with Equations (18) and (19) yield the following theorem.

**Theorem 3.** *Let $\epsilon > 0$. If the Hessian and the Lipscitz constant of $f(z)$ is bounded, then with probability at least $1 - 8\delta$,*

$$R_q(\hat{g}) \leq R_q(g_q^*) + c|L|\left(\left(\sqrt{V} + \sqrt{\log\frac{1}{\delta}}\right)\left(\left(\frac{\log(n/\delta)}{n}\right)^{1/3} + \sqrt{\frac{4^d \log(n/\delta) + o_n(1/\epsilon)}{n}}\right) + G(\epsilon)\right),$$
$$\tag{22}$$

*where $c$ is a universal constant and $\hat{g}$ is the minimizer in Step 6 of Algorithm 2.*

### B.1 Proof of Lemma 2

We need few definitions to prove Lemma 2. For a point $z$, let $B_n(z)$ be a ball around it such that

$$\Pr_{Z \sim f(z)}(Z \in B_n(z)) = \frac{\log \frac{n^2}{\delta}}{n} \triangleq \alpha_n.$$

Intuitively, with high probability the nearest neighbor of each sample $z$ lies in $B_n(z)$. We formalize it in the next lemma.

**Lemma 3.** *With probability $> 1 - \delta$, the nearest neighbor of each sample $z \in \mathcal{Z}_2$ in $\mathcal{Z}_3$ lie in $B(z)$.*

*Proof.* The probability that none of $\mathcal{Z}_3$ appears in $B(z)$ is $1 - (1 - \alpha_n)^n \leq \delta/n$. The lemma follows by the union bound. $\square$

We now bound the probability that the the nearest neighbor balls $B_n()$ intersect for two samples.

**Lemma 4.** *Let $\epsilon > 0$. If the Hessian of the density ($f(z)$) is bounded by $c$ and the Lipschitz constant is bounded by $\gamma$, then for any given $z_1$ such that $f(z_1) \geq \epsilon$ and a sample $z_2 \sim f$,*

$$\Pr_{z_2 \sim f}(B_n(z_1) \cap B_n(z_2) \neq \emptyset) \leq \beta_n \triangleq 4^d \alpha_n(1 + o_n(1/\epsilon)).$$

*Proof.* Let $r_n(z)$ denote the radius of $B_n(z)$. Let $B(z, r)$ denote the ball of radius $r$ around $z$ and $V(z, r)$ be its volume. We can rewrite $\beta_n$ as

$$= \Pr(B_n(z_1) \cap B_n(z_2) \neq \emptyset)$$
$$= \Pr(B_n(z_1) \cap B_n(z_2) \neq \emptyset, 3r_n(z_1) \geq r_n(z_2)) + \Pr(B_n(z_1) \cap B_n(z_2) \neq \emptyset, 3r_n(z_1) < r_n(z_2)).$$

We first bound the first term. Note that

$$\int_{z' \in B_n(z)} f(z')dz' = \alpha_n,$$

Hence, by Taylor's series expansion and the bound on Hessian yields,

$$\alpha_n = \int_{z' \in B_n(z)} f(z')dz' = V(z, r_n(z)) \left( f(z) + O(r_n^2(z)c) \right).$$

Similarly,

$$\Pr(z' \in V(z, 4r_n(z))) = V(z, 4r_n(z)) \left( f(z) + O(9r_n^2(z)c) \right)$$
$$= 4^d \alpha_n (1 + o_n(1/\epsilon)),$$

where the last equality follows from the fact that $V(z, 4r_n(z))/V(z, r_n(z)) = 4^d$ in $d$ dimensions.

Then the first term can be bounded as

$$\Pr(B_n(z_1) \cap B_n(z_2) \neq \emptyset, 3r_n(z_1) \geq r_n(z_2)) = \Pr(z_2 \in B(z_1, r_n(z_1) + r_n(z_2)), 3r_n(z_1) \geq r_n(z_2))$$
$$\leq \Pr(z_2 \in B(z_1, 4r_n(z_1)), 3r_n(z_1) \geq r_n(z_2))$$
$$\leq \Pr(z_2 \in B(z_1, 4r_n(z_1)))$$
$$\leq 4^d \alpha_n (1 + o_n(1/\epsilon)).$$

To bound the second term, observe that if $B_n(z_1) \cap B_n(z_2) \neq \emptyset$ and $3r_n(z_1) < r_n(z_2)$. There exists a point $z'$ on the line joining $z_1$ and $z_2$ at distance $3r_n(z_1)$ from $z_1$ such that

$$\Pr(z'' \in B(z', 3r_n(z_1))) < \alpha_n.$$

As before bound on the Hessian yields,

$$\alpha_n > V(z', 3r_n(z_1))(f(z') - O(9r_n^2(z_1)c)).$$

Hence,

$$f(z') < 3^{-d}(f(z) + O(r_n^2(z)c)) + O(9r_n^2(z_1)c).$$

However, $f(z) > \epsilon$ and $f(z') \geq f(z) - 3r_n(z_1)\gamma$ and $r_n(z_1) \to 0$. Hence, a contradiction. Thus

$$\Pr(B_n(z_1) \cap B_n(z_2) \neq \emptyset, 3r_n(z_1) < r_n(z_2)) = 0.$$

$\square$

Consider the graph on indices $[n]$, such that two indices are connected if and only if $B_n(z_i) \cap B_n(z_j) \neq \emptyset$, $f(z_1) \geq \epsilon$, $f(z_2) \geq \epsilon$. Let $\Delta(Z_1^n)$ be the maximum degree of the resulting graph. We first show that the maximum degree of this graph is small.

**Lemma 5.** *With probability* $\geq 1 - \delta$,

$$\Delta(Z_1^n) \leq 4n\beta_n.$$

*Proof.* For index 1, by Lemma 4 that probability of $j$ points intersect is at most

$$\sum_{i=0}^{j} \binom{n}{i} \beta_n^i (1 - \beta_n)^{n-i}.$$

Hence, the degree of vertex 1 is dominated by a binomial distribution with parameters $n$ and $\beta_n$. The lemma follows from the union bound and the Chernoff bound. $\square$

Let $k > 2\Delta(Z_1^n)$ and $S_0, S_1, S_2 \dots S_k$ be $k$ independent sets of the above graph such that $\max_{t \geq 1} |S_t| \leq 2n/k$. Note that such independent sets exists by Lemma 10. We set the exact value of $k$ later. Let $S_0$ contains all indices such that $f(z_i) < \epsilon$.

**Lemma 6.** *With probability* $> 1 - \delta$,

$$|S_0| \leq nG(\epsilon) + \sqrt{n \log \frac{1}{\delta}}.$$

*Proof.* Observe that $|S_0|$ is the sum of $n$ independent random variables and changing any of them changes $S_0$ by at most 1. The lemma follows by McDiarmid's inequality. □

We can upper bound the LHS in Equation (21) as

$$\sup_{g \in \mathcal{G}} (R_S(g) - R_n(g|\mathcal{Z})) \leq \sum_{t=1}^{k} \frac{|S_t|}{n} \sup_{g \in \mathcal{G}} \frac{1}{|S_t|} \sum_{i \in S_t} (L(g(u_i), \ell_i) - h(Z_i)) + \frac{|S_0|}{n} |L|.$$

Let $N(Z_i)$ denote the number of elements of $\mathcal{Z}_3$ that are in $B(Z_i)$ and Let $A_i$ be always true if $Z_i \in \mathcal{Z}_1$ and otherwise $A_i$ be the event such that nearest neighbor of samples in $N(Z_i) > 0$. We first show the following inequality.

**Lemma 7.** *With probability* $\geq 1 - \delta$, *for all sets* $S_t$.

$$\sup_{g \in \mathcal{G}} \frac{1}{|S_t|} \sum_{i \in S_t} (L(g(u_i), \ell_i) - h(Z_i)) = \sup_{g \in \mathcal{G}} \frac{1}{|S_t|} \sum_{i \in S_t} 1_{A_i} (L(g(u_i), \ell_i) - h(Z_i)).$$

*Proof.* Let $X_i = (L(g(u_i), \ell_i) - h(Z_i))$ and Observe that LHS can be written as

$$\sup_{g \in \mathcal{G}} \frac{1}{|S_t|} \sum_{i \in S_t} X_i = \sup_{g \in \mathcal{G}} \frac{1}{|S_t|} \sum_{i \in S_t} X_i 1_{A_i} + \sup_{g \in \mathcal{G}} \frac{1}{|S_t|} \sum_{i \in S_t} (X_i - X_i 1_{A_i}).$$

If the conditions of Lemma 3 hold, the nearest sample of $Z_i$'s lie within $B_n(Z_i)$. Hence, with probability $\geq 1 - \delta$, the second term is 0. Hence the lemma. □

Let $r$ be defined as follows.

$$r(Z_i, N_i) \triangleq \mathbb{E}[1_{A_i} (L(g(u_i), \ell_i) - h(Z_i)) | N(Z_i) = N_i].$$

Observe that

$$\mathbb{E}\left[ \sum_{i \in S_t} 1_{A_i} (L(g(u_i), \ell_i) - h(Z_i)) | N(Z_1), \dots N(Z_n) \right]$$

$$= \sum_{i \in S_t} \mathbb{E}\left[ 1_{A_i} (L(g(u_i), \ell_i) - h(Z_i)) | N(Z_1), \dots N(Z_n) \right]$$

$$= \sum_{i \in S_t} \mathbb{E}\left[ 1_{A_i} (L(g(u_i), \ell_i) - h(Z_i)) | N(Z_i) \right]$$

$$= \sum_{i \in S_t} r(Z_i, N_i). \tag{23}$$

Hence, we can split the term as

$$\sup_{g \in \mathcal{G}} \frac{1}{|S_t|} \sum_{i \in S_t} 1_{A_i} (L(g(u_i), \ell_i) - h(Z_i))$$

$$\leq \sup_{g \in \mathcal{G}} \frac{1}{|S_t|} \sum_{i \in S_t} 1_{A_i} (L(g(u_i), \ell_i) - r(Z_i, N(Z_i))) + \sup_{g \in \mathcal{G}} \frac{1}{|S_t|} \sum_{i \in S_t} 1_{A_i} (r(Z_i, N(Z_i)) - h(Z_i))$$

$$\tag{24}$$

Given $\{Z_i, N(Z_i)\}$, the first term in the RHS of the Equation (24) is a function of $|S_t|$ independent random variables as $1_{A_i} * L(g(u_i), \ell_i)$ are mutually independent given $\{Z_i, N(Z_i)\}$. Thus we can

use standard tools from VC dimension theory and state that with probability $\geq 1 - \delta$, the first term in the RHS of Equation (24) can be upper bounded as

$$\sup_{g \in \mathcal{G}} \frac{1}{|S_t|} \sum_{i \in S_t} 1_{A_i} \left( L(g(u_i), \ell_i) - r(Z_i, N(Z_i)) \right) \leq |L| C \sqrt{\frac{V}{|S_t|}} + |L| \sqrt{\frac{\log(1/\delta)}{|S_t|}}.$$

conditioned on $\{Z_i, N(Z_i)\}$

To bound the second term in the RHS of Equation (24), observe that unlike the first term, the $N(Z_i)$s are dependent on each other. However note that $N(Z_1), \ldots N(Z_{|S_t|})$ are distributed according to multinomial distribution with parameters $n$ and $\alpha_n$. However, if we replace them by independent Poisson distributed $N(Z_i)$s we expect the value not to change. We formalize it by total variation distance. By Lemma 9, the total variation distance between a multinomial distribution and product of Poisson distributions is

$$\mathcal{O}(|S_t|\alpha_n),$$

and hence any bound holds in the second distribution holds in the first one with an additional penalty of

$$\mathcal{O}\left(|S_t|\alpha_n|L|\right).$$

Under the new independent sampling distribution, again the samples are independent and we can use standard tools from VC dimension and hence, with probability $\geq 1 - \delta$, the term is upper bounded by

$$|L| C \sqrt{\frac{V}{|S_t|}} + |L| \sqrt{\frac{\log(1/\delta)}{|S_t|}}.$$

Hence, summing over all the bounds, we get

$$\sup_{g \in \mathcal{G}} (R_S(g) - R_n(g|\mathcal{Z})) \leq |L| \mathcal{O} \left( \frac{|S_0|}{n} + \sum_{t=1}^{k} \frac{|S_t|}{n} \left( \sqrt{\frac{\log \frac{1}{\delta}}{|S_t|}} + c\sqrt{\frac{V}{|S_t|}} + \alpha_n |S_t| \right) \right)$$

$$\leq |L| \mathcal{O} \left( G(\epsilon) + \sqrt{\frac{k}{n}} \left( \sqrt{\log \frac{1}{\delta}} + c\sqrt{V} \right) + \alpha_n \max_t |S_t| + \sqrt{\frac{\log \frac{1}{\delta}}{n}} \right)$$

$$\leq |L| \mathcal{O} \left( \sqrt{\frac{k}{n}} \left( \sqrt{\log \frac{1}{\delta}} + c\sqrt{V} \right) + \alpha_n \frac{n}{k} + \sqrt{\frac{\log \frac{1}{\delta}}{n}} \right).$$

conditioned on $Z_i, N(Z_i)$ Choose $k = n\alpha_n^{2/3} + 8n\beta_n$, and note that the conditioning can be removed as the term on the r.h.s are constants. This yields the result. The error probability follows by the union bound.

**Theorem 4.** *Assume the conditions for Theorem 3. Suppose the loss is $L(g(u), \ell) = 1_{g(u) \neq \ell}$ (s.t $|L| \leq 1$). Further suppose the class of classifying function is such that $R_q(g_q^*) \leq r_0 + \eta$. Here, $r_0 \triangleq 0.5(1 - d_{TV}(q(x, y, z|1), q(x, y, z|0)))$ is the risk of the Bayes optimal classifier when $\mathbb{P}(\ell = 1) = \mathbb{P}(\ell = 0)$. This is the best loss that any classifier can achieve for this classification problem [4]. Under this setting, w.p at least $1 - 8\delta$ we have:*

$$\frac{1}{2} \left( 1 - d_{TV}(f, f^{CI}) \right) - \frac{b(n)}{2} \leq R_q(g_S) \leq \frac{1}{2} \left( 1 - d_{TV}(f, f^{CI}) \right) + \frac{b(n)}{2} + \eta + \gamma_n$$

*Proof.* Assume the bounds of Theorem 3 holds which happens w.p at least $1 - 8\delta$. From Theorem 3 we have that

$$R_q(g_S) \leq R_q(g_q^*) + \gamma_n. \tag{25}$$

Also, note that from Theorem 1 we have the following:

$$d_{TV}(q(x, y, z|1), q(x, y, z|0)) = d_{TV}(\phi, f)$$
$$\leq d_{TV}(\phi, f^{CI}) + d_{TV}(f^{CI}, f)$$
$$\leq b(n) + d_{TV}(f^{CI}, f) \tag{26}$$

Under our assumption we have $R_q(g_q^*) \leq r_0 + \eta$. Combining this with (25) and (26) we get the r.h.s. For, the l.hs note that $R_q(g_q^*) \geq r_0$ as the bayes optimal classifier has the lowest risk. We can now use (26) to prove the l.h.s. $\qquad \square$

## C   Tools from probability and graph theory

**Lemma 8** (McDiarmid's inequality [20])**.** *Let $X_1, X_2, \ldots X_m$ be $m$ independent random variables and $f$ be a function from $x_1^n \to \mathbb{R}$ such that changing any one of the $X_i$s changes the function $f$ at most by $c_i$, then*

$$\Pr(f - \mathbb{E}[f] \geq \epsilon) \leq \exp\left(\frac{-2\epsilon^2}{\sum_{i=1}^m c_i^2}\right).$$

**Lemma 9** (Special case of Theorem 1 in [25])**.** *Let $f_m$ be the multinomial distribution with parameters $n$ and $p_1, p_2, \ldots p_k, 1 - \sum_{i=1}^k p_i$, and $f_p$ be the product of Poisson distributions with mean $np_i$ for $i \leq 1 \leq k$, then*

$$d_{TV}(f_m, f_s) \leq 8.8 \sum_{i=1}^k p_i.$$

**Lemma 10.** *For a graph with maximum degree $\Delta$, there exists a set of independent sets $S_1, S_2, \ldots S_k$ such that $k \geq 2\Delta$ and*

$$\max_{1 \leq i \leq k} |S_i| \leq 2n/k.$$

*Proof.* We show that the following algorithm yields a coloring (and hence independent sets) with the required property.

Let $1, 2, \ldots k$ be $k$ colors, where $k > 2\Delta$. We arbitrarily order the nodes, and sequentially color nodes with a currently least used color from among the ones not used by its neighbors. Consider the point in time when $i$ nodes have been colored, and we evaluate the options for the $(i + 1)^{th}$ node. The number of possible choices of color for that node is $c \geq k - \Delta$. Out these $c$ colors, the average number of nodes belonging to each color at this point is at-most $i/c$. Therefore by pigeonholing, the minimum is less than the average; thus the number of nodes belonging to chosen color is no larger than $i/c \leq i/(k - \Delta)$.

Hence at the end when all $n$ nodes are colored, each color has been used no more than $(n - 1)/(k - \Delta) + 1 < 2n/k$.   $\square$