[Reviews · NeurIPS 2017]

Reviewer 1



This paper proposed a model powered approach to conduct conditional independent tests for iid data. The basic idea is to use nearest neighbor bootstrap to generate samples which follow a distribution close to the f^{CI} and a classifier is trained and tested to see if it is able to distinguish the observation distribution and the nearest neighbor bootstrapped distribution. If the classification performance is close to the random guess, one fails to reject the null hypothesis that data follows conditional independence otherwise one accept the null hypothesis. Authors developped some bounds on the closeness of the nn bootstrapped distribution and f^{CI} in term of total variantional distance and bounds on empirical risks under ideal classification settings and for near-Independent samples. In general, the paper is trying to address an important problem and the paper is presented in a clear way. Detail comments: Major: 1. It seems that the whole method can be decoupled into two major components. One is to generate samples that mimic the f_{CI} and the other one is to use a classifier for determination. My question is whether we can replace any one step by existing solutionsfrom literature. For example, we used the permutation based method proposed in [7] and then the classification based approach to solve the problem. Or we use the nn bootstrap and the kernel two sample test approach together to solve the problem. I am couriois about the performances. 2. The nearest neighbor bootstrap distribution is close to f_{CI} in the finite sample size setting. In this case, if the groundtruth is x and y are weakly dependent given z, how the proposed method is going to tell? 3. Is the method totally symmetric with respect to x and y? In the causal perspective, x and y can be dependent given z due to two different causal models: x causes y or y causes x. In this two different scenarios, shall we need to determine which variable to sample. Authors did not consider this problem. 4.I am seriously concerned about how to choose the parameter $\tau$ in algorithms 2 and 3. Minor: In algorithm 2 and 3, it seems that the empirical risk should be devided by the sample size.

Reviewer 2



The paper proposes a new conditional independence test. Strengths 1) The paper identifies that in ordered for a test to work an assumption on the smoothness of conditional density should be made. Authors propose such an assumption and bound distance in total variation between density under the null hypothesis and density obtained from bootstrap under this assumption. This is novel. 2) Paper uses this assumption to bound error of the optimal classifier on the training set (1.1 (iii)) 3) Theorem 1, which combines the assumption on smoothness, the assumption that hat R can be achieved with error eta and the assumption on difference in errors between optimal and trained classifier, is novel, insightful and non-trivial. Weaknesses Paper is hard to read and does not give much interpretation of the results, to an extent that I am not sure if I understand them correctly. E.g. consider inequality in the section 1.1 iii) assume the alternative hypothesis holds and r_0=0.1. Suppose G is a small class, say linear functions, and suppose they are not expressive enough to distinguish between f and f^CI at all. Can you explain, intuitively, how the small probability of the error can be achieved ( hat R < 0.1 + o_1 < 0.2 for large n)?

Reviewer 3



The authors propose a general test to test conditional independence between three continuous variables. The authors don't have strong parametric assumptions -- rather, they propose a method that uses samples from the joint distribution to create samples that are close to the conditional product distribution in total variation distance. Then, they feed these samples to a binary classifier such as a deep neural net, and after training it on samples from each of the two distributions, use the predictive error of the net to either accept or reject the null hypothesis of conditional independence. They compare their proposed method to those existing in literature, both with descriptions and with experimental results on synthetic and real data. Conditional independence testing is an important tool for modern machine learning, especially in the domain of causal inference, where we are frequently curious about claims of unconfoundedness. The proposed method appears straightforward and powerful enough to be used for many causal applications. Additionally, the paper is well-written. While the details of the theoretical results and proofs can be hard to follow, the authors do a good job of summarizing the results and making the methodology clear to practitioners. To me, the most impressive part of the paper is the theoretical results. The proposed method is a somewhat straightforward extension of the method for testing independence in the paper "Revisiting Classifier Two-Sample Tests" (https://arxiv.org/pdf/1610.06545.pdf), as the authors acknowledge (they modify the test to include a third variable Z, and after matching neighbors of Z, the methods are similar). However, the guarantees of the theoretical results are impressive. A few more comments: -Is there a discrete analog? The bootstrap becomes even more powerful when Z is discrete, right? -How sensitive is the test on the classification method used? Is it possible that a few changed hyperparameters can lead to different acceptance and rejection results? This would be useful to see alongside the simulations -Section 1.1 of Main Contributions is redundant -- the authors summarize their approach multiple times, and the extent appears unnecessary. -Writing out Algorithm 2 seems unnecessary — this process can be described more succinctly, and has been described already in the paper -Slight typos — should be “classifier” in line 157, “graphs” in 280